# Influence of Hypoxic and Hyperoxic Preconditioning on Endothelial Function in a Model of Myocardial Ischemia-Reperfusion Injury with Cardiopulmonary Bypass (Experimental Study)

**DOI:** 10.3390/ijms21155336

**Published:** 2020-07-27

**Authors:** Irina A. Mandel, Yuri K. Podoksenov, Irina V. Suhodolo, Darya A. An, Sergey L. Mikheev, Andrey Yu. Podoksenov, Yulia S. Svirko, Anna M. Gusakova, Vladimir M. Shipulin, Andrey G. Yavorovskiy

**Affiliations:** 1Tomsk National Research Medical Center of the Russian Academy of Sciences, Cardiology Research Institute, 111a Kievskaya Str., Tomsk 634012, Russia; uk@cardio-tomsk.ru (Y.K.P.); msl1912@mail.ru (S.L.M.); Paucardio1977@mail.ru (A.Y.P.); julia@cardio-tomsk.ru (Y.S.S.); mag_a@mail.ru (A.M.G.); shipulin@cardio-tomsk.ru (V.M.S.); 2I.M. Sechenov First Moscow State Medical University of the Ministry of Health of the Russian Federation (Sechenov University), 8/2 Trubetskaya Str., Moscow 119991, Russia; yavor@bk.ru; 3Federal Scientific and Clinical Center of Specialized Types of Medical Care and Medical Technologies of the Federal Medical and Biological Agency of Russia, 28 Orekhoviy Blvd., Moscow 115682, Russia; 4Siberian State Medical University of the Ministry of Health of the Russian Federation, 2 Moskovskiy Tract Str., Tomsk 634050, Russia; staranie@mail.ru (I.V.S.); darya.an.1996@gmail.com (D.A.A.); 5Swiss Medica XXI C.A., 21/1 Annenskaya str., Moscow 127521, Russia

**Keywords:** hypoxic preconditioning, hyperoxic preconditioning, ischemia-reperfusion injury, endothelial dysfunction, nitric oxide, endothelin-1, cardiopulmonary bypass, experiment

## Abstract

The aim of the experiment was to evaluate the effect of preconditioning based on changes in inspiratory oxygen fraction on endothelial function in the model of ischemia-reperfusion injury of the myocardium in the condition of cardiopulmonary bypass. The prospective randomized study included 32 rabbits divided into four groups: hypoxic preconditioning, hyperoxic preconditioning, hypoxic-hyperoxic preconditioning, and control group. All animals were anesthetized and mechanically ventilated. We provided preconditioning, then started cardiopulmonary bypass, followed by induced acute myocardial infarction (ischemia 45 min, reperfusion 120 min). We investigated endothelin-1, nitric oxide metabolites, asymmetric dimethylarginine during cardiopulmonary bypass: before ischemia, after ischemia, and after reperfusion. We performed light microscopy of myocardium, kidney, lungs, and gut mucosa. The endothelin-1 level was much higher in the control group than in all preconditioning groups after ischemia. The endothelin-1 even further increased after reperfusion. The total concentration of nitric oxide metabolites was significantly higher after all types of preconditioning compared with the control group. The light microscopy of the myocardium and other organs revealed a diminished damage extent in the hypoxic-hyperoxic preconditioning group as compared to the control group. Hypoxic-hyperoxic preconditioning helps to maintain the balance of nitric oxide metabolites, reduces endothelin-1 hyperproduction, and enforces organ protection.

## 1. Introduction

Coronary artery bypass grafting (CABG) with cardiopulmonary bypass is a common surgical therapy for patients suffering from coronary artery diseases. During surgery, the heart is subjected to a long period of ischemia due to the occlusion of aortic artery. The heavy burden of myocardial ischemia-reperfusion injury (IRI) thus induces cardiomyocyte death, which can paradoxically reduce the beneficial effect of CABG [1]. Acute cardiovascular dysfunction occurs perioperatively in more than 20% of cardiosurgical patients [2]. Twenty-five percent of patients undergoing elective CABG surgery require inotropic support for postoperative myocardial dysfunction. Acute heart failure associated with a failure to wean patients off cardiopulmonary bypass (CPB) may be surgery related, patient-specific, or both [3]. CABG surgery is associated with systemic inflammatory response, endothelial damage, and platelet activation regardless of the use of CPB. Eight percent of patients developed postoperative myocardial infarction after the CABG with CPB, 6.3% developed low cardiac output syndrome, and 3.7% patients died during the early postoperative period due to extensive postoperative myocardial infarction [4]. The systemic inflammatory reaction syndrome is manifested during nonpulsatile mode of CPB. Not only myocardium but other organs are affected with hypoperfusion and inflammation during CBP. The preconditioning methods are able to improve or prevent the organs damage during CPB.

Ischemic preconditioning was first described by C.E. Murry and colleagues as the cardioprotective mechanism that are activated by brief ischemic stress. These mechanisms can reduce the rate of cardiac cell necrosis during a subsequent episode of prolonged ischemia. It is now evident that this endogenous myocardial protective mechanism can be activated by either coronary occlusion or hypoxia. Mechanisms implicated in ischemic preconditioning include signalling reactive oxygen species (ROS), nitric oxide (NO), attenuated formation of detrimental ROS, reduced endothelin-1, enhanced endothelial NO-synthase function, and preservation of endothelial tight junctions [5,6]. Hypoxia or hyperoxia could be such a stimulus that will induce myocardial endogenous protective properties. The mechanisms of hypoxic and hyperoxic preconditioning partially have common ways of implementing a protective effect. ROS generated by different types of preconditioning and postconditioning are known to act as triggers of cardiac protection [7,8,9,10].

Preconditioning by moderate hypoxia or hyperoxia serves as an effective drug-free method to increase the organism resistance to negative effects including IRI [11,12,13,14,15]. It has been firmly established that the diminished oxygen delivery to the tissues in response to hypoxia is countered by a combination of the increased regional blood flow and the enhanced functional capillary density in the microcirculation [16]. A recent study has pointed out an important role of hypoxia signaling in the protection from organ injury, including myocardial infarction, acute respiratory distress syndrome, and acute kidney or gut injury [17,18,19].

M. Rocco et al. described the phenomenon of the “normobaric oxygen paradox”: relative hypoxia induced after a hyperoxia period acts as a hypoxic trigger able to significantly increase the erythropoietin or hemoglobin levels [20]. In experimental studies, exposure to hyperoxia for a limited time before ischemia induces a low-grade systemic oxidative stress, evokes a preconditioning-like effect of the myocardium, and reduces the infarction area by 20% and the number of arrhythmias after ischemia-reperfusion [7,9,10,18,21,22]. The interval hypoxic-hyperoxic training increases the exercise tolerance of patients with ischemic heart disease [23].

Endothelial dysfunction during IRI is associated with a decrease in nitric oxide (NO) production, endothelial damage-causing dysregulation of vascular tone, and prolonged vasoconstriction. The key role of endothelin-1 (ET-1) is determined in the regulation of remodelling and myocardial hypertrophy, angiogenesis, extracellular matrix production, as well as the excretion of sodium and water [24]. The increased ET-1 level positively correlates with the severity of necrotic injury and with the incidence of arrhythmias in acute myocardial infarction [25]. The ET-1 level significantly increased during the CABG surgery with CPB [26].

Previous studies clearly demonstrated that the deficiency of endothelial NO-synthase exacerbates myocardial IRI, whereas its overexpression, the administration of NO donors, and the inhaled NO, all protect the myocardium [27,28,29]. NO is an important key mediator and trigger in cardio protection afforded by both early and late ischemic preconditioning [29]. NO is one of the key links in pathophysiology of oxidative stress and, in particular, in the process of ischemia and reperfusion in cardiac surgery by using CPB [30].

The asymmetric dimethylarginine (ADMA) decreases the intracellular NO production [31]. The ADMA concentration in plasma is a biomarker for endothelial dysfunction associated with the increased risk of cardiovascular mortality [32].

Many studies point to the disruption of the NO production and to ET-1 hyperproduction in the CABG with CPB, however the studies are contradictory [33,34].

The aim of the experiment was to estimate the effect of preconditioning based on changes in inspiratory oxygen fraction on endothelial function in the model of IRI of the myocardium in the condition of CPB. We hypothesize that the preconditioning with a different fraction of inspired oxygen modulates the function of the endothelium, and implements the protective properties by maintaining the concentration of NO metabolites.

## 2. Results

### 2.1. Hemodynamic Data During the Experiment

Statistical analysis of the baseline data didn’t reveal any differences between the groups. Animals of each group were subjected to identical experimental procedures. Aerobic cardiac function prior to ischemia was unaffected by the treatment with hypoxia and/or hyperoxia (Table 1). The heart was beating during the CPB. The ischemia and reperfusion periods demonstrated different types of ventricular arrhythmias, that did not affect hemodynamics. We observed minimal hemodilution after the CPB beginning due to the prime volume (100 mL of crystalloid solution in each case). There were no changes in central temperature (Table 2). The oxygen haemoglobin saturation matched the inhaled oxygen fraction (hypoxic period pulsoxymetry 86 ± 2%, hyperoxic period pulsoxymetry 100%). Thus, there were no changes in the pre-ischemic function as a consequence of the hypoxic/hyperoxic treatment that were predictive of the post-ischemic recovery.

### 2.2. Metabolic State During Experiment

The SvO_2_, O_2_EI, Pcv-aCO_2_/Ca-vO_2_, lactate, and glucose were within the reference limits during the experiment for all compared animals, which indicates the stability of the metabolic processes. There was a regular change of SaO_2_, which confirmed the target oxygen concentration in the respiratory gas at the level of 10% and 75–80% (Table 3).

### 2.3. The Dynamics of Endothelin-1 Concentration in Blood Plasma During Experiment

The ET-1 level increased after ischemia (*p* = 0.003) and after the reperfusion (p=0.005) in all groups compared to the baseline in each group. Pairwise comparison showed the highest increase in ET-1 concentration in the control group of animals. The ET-1 level after ischemia was as high as 3.466 [2.401; 3.893] fmol/mL in control group in comparison with 1.931 [1.730; 2.395] fmol/mL in the Hypoxic-hyperoxic preconditioning (HHP) Group (*p* = 0.006). The ET-1 level was 6.199 [5.113; 6.846] fmol/mL in the control group, and 3.166 [2.351; 4.388] fmol/mL (*p* = 0.003) in the HHP group after the reperfusion (Figure 1).

### 2.4. Nitrite Production

The nitrite concentration after ischemia and reperfusion was significantly lower in all groups in comparison with the baseline (Table 4). Although the nitrite level in the hypoxic preconditioning (HypP), hyperoxic preconditioning (HyperP), and HHP groups decreased relative to the baseline at the stages after ischemia and reperfusion, it was significantly higher than that in the Control Group.

The intragroup assessment of the nitrite concentration dynamics in the HypP, HHP, and control groups revealed a significant decrease of this indicator in response to myocardial ischemia-reperfusion relative to the baseline (*p* = 0.012). In the HyperP group, the nitrite concentration decreased minimally in comparison with the other groups.

### 2.5. Nitrate Production

The intragroup dynamics of nitrate concentration showed a decrease in the HypP, HyperP, and control groups at the stages after ischemia and reperfusion in comparison with the baseline (before ischemia, *p* = 0.012), as shown in Table 4. The dynamics of nitrate concentration in the HHP group were minimal during the experiment (*p* = 0.069).

The pairwise comparison of nitrate concentration at the stage before ischemia and after ischemia revealed a significant difference between HypP and HyperP against the control group. The nitrate level after the reperfusion differed between the HyperP and the control group (*p* = 0.012) and between the HHP and the control group (*p* = 0.027).

### 2.6. Total Concentration of Nitric Oxide Metabolites

The analysis of the total concentration of nitric oxide metabolites (NOx.total) revealed a difference between the groups at the baseline (after preconditioning and before ischemia, *p* = 0.034), after ischemia (p=0.014), and after reperfusion (*p* = 0.022), as shown in Figure 2. The level of NOx.total at the stage before ischemia in the HypP group was 15.975 [13.709; 23.988] µmol/L. In the HyperP group, it was 18.694 [17.101; 26.916] µmol/L, in the HHP group it was 16.625 [11.072; 26.000] µmol/L, and in the control group, it was 11.151 [9.130; 14.572] µmol/L. After ischemia, the NOx.total level differed in the HypP group (*p* = 0.009), HyperP group (*p* = 0.005), and the HHP group (*p* = 0.027) compared with the control group. After the reperfusion The NOx.total level in the experimental groups was also significantly higher than that in the control group after the reperfusion.

### 2.7. ADMA Production

The analysis of the ADMA concentration revealed a significant difference between the HHP and the HyperP groups against the control group at the stage after ischemia (Table 4). The ADMA level increased after ischemia and reperfusion in the control group only.

### 2.8. Morphology

Light microscopy revealed disturbances of myocardial structure, increase in the distance between the discs, and a lack of cross-striations in the hibernation area. We observed less intense damage in HHP animals compared with the control group (Figure 3). We found reversible (segmental contractions of I and II degree) and irreversible (disintegration of myofibrils, segmental contracture of III degree) changes in myofibrils. Predominantly reversible changes were found in the HHP group. Light microscopy of kidneys revealed marked edema of cortical and medullar substances in the control group (Figure 4). Gut mucosa and lungs had enlarged capillaries, sometimes filled with erythrocytes (Figure 5 and Figure 6). Light microscopy of gut mucosa, lung parenchyma, and kidneys were less affected in the HHP group.

### 2.9. Morphometry

We presented data from two groups (HHP and Control), as the biggest differences were revealed in them. A statistically significant decrease of 40% in the volume of perinuclear vacuolization was detected in the ischemic area of the myocardium of the HHP group compared to the control group. An increase in the volume of cardiomyocytes by 12%, a decrease in the volume of interstitial edema, and a decrease in the volume of perinuclear vacuolization by 46% and vessels by 41% were detected in the area at risk of the HHP group (Table 5).

The volume of ischemic cardiomyocytes in the HHP group was statistically less significantly than that in the control group. The volume of non-ischemic cardiomyocytes in HHP groups was greater than in the control group (Table 6).

## 3. Discussion

The major findings of our study comprise of a consistently higher level of ET-1 after ischemia and reperfusion in all groups in comparison to the baseline. The highest increase of ET-1 level was observed in animals of the control group after reperfusion. This factor was 1.5–2-fold lower in preconditioned animals. The nature of change in the ET-1 concentration indicates the modulating effect of preconditioning on endothelial function in the form of elimination ET-1 hyperproduction in response to stress (IRI). The ET-1 is synthesized very quickly under the influence of many factors: adrenaline, angiotensin-II, hypoxia, cytokines, IRI, and CPB [24,35,36,37]. The adaptive role of ET-1 hyperproduction in patients with acute heart failure in the post-operation period is maintaining the perfusion of vitals due to inotropic effect. Perhaps the ET-1 hyperproduction thus led to microcirculatory dysfunction due to strong vasoconstriction activity.

The second important finding was the greater level of the total concentration of nitric oxide metabolites in the preconditioned animals after IRI. During the experiment, the concentration of NO metabolites decreased in all groups. However, in the preconditioning groups its concentration exceeded the control group by almost twice. This data coincides with data of B.P. Cabigas (2006), who suggested that nitrite and nitrate production was increased more than two-fold in rats treated with hyperoxia [38].

The nitrite and nitrate levels before ischemia but after precondition was significantly higher than that in the control group. Although the nitrite level in the preconditioned animals decreased after ischemia relative to the baseline in each group, it was significantly higher than that in the control group. Such dynamics were revealed in patients exposed to CABG on the background of non-pulsing CBP but not in patients who came through surgery on a beating heart [33]. According to our data, the concentrations of nitrite and total nitric oxide metabolites were higher, the nitrate level was stable, and the ADMA level was lesser at all stages in the HyperP and HHP groups. Consequently, hyperoxia might have led to the stability of nitrite production in stressful conditions.

The third finding was the ADMA level elevation which could be associated with reductions of NO especially in the control group and in HypP. Our data were similar to those of Stühlinger et al. (2007) [39]. The comparison of the ADMA level between the HHP, HyperP, and the control group demonstrated a significant difference after ischemia. This fact is consistent with a more stable level of nitrite concentration and a higher nitrate level in the HHP group at all stages of the experiment. We observed a lesser effect of HypP probably due to short exposure time, unlike many other studies in the hypoxic preconditioning [12,15,16,18].

Vessel training in the form of intermittent vasoconstriction and vasodilation as a result of hypoxic and hyperoxic influences may constitute a possible mechanism of protection for cardiomyocytes and cells of other organs. The action mechanisms of hypoxic and hyperoxic preconditioning partially have common ways of implementing a protective effect. It is known that ROS plays a triggering role in the ischemic and hypoxic preconditioning, pharmacological, and remote ischemic preconditioning [7,40].

In Vignon-Zellweger et al. (2012) review of endothelin and endothelin receptors in the renal and cardiovascular systems clearly described the mechanisms of vasoconstriction and vasodilatation. ET-1 is a strong vasoconstrictor induces the production of vasodilator NO. ET-1 is also responsible for the positive inotropic effect of angiotensin II via the production of reactive oxygen species. ET-1 is overexpressed in the failing heart but may prevent apoptosis and restore cardiac function in stress situation [41]. In our study, we observed the vasoconstriction and vasodilatation effects of changes in ET-1 and NO production during stress (ischemia-reperfusion injury) on the background of preconditioning or without it. So, we hypothesize that changes in vascular tone are the basis of vascular training before stress situation.

There are now thousands of studies which reported >100 different signaling molecules and mechanisms of conditioning in a wide range of experimental preparations, from isolated subcellular structures to isolated cardiomyocytes with exposure to hypoxia/reoxygenation, from cultured cardiomyoblast cell lines to freshly isolated neonatal or adult cardiomyocytes, from buffer-perfused isolated hearts with global or regional ischemia and reperfusion to in vivo and in situ preparations with coronary artery occlusion and reperfusion, using different methodologies from biochemical to immunoblotting techniques [42]. Intensive research has been performed to elucidate the signalling pathways, which become activated upon a conditioning stimulus [43]. The protein kinase B, endothelial nitric oxygen synthase, glycogen synthase kinase 3b, p44/42, and signal transducer and activator of transcription 3 may be considered as indicators of the intracellular changes taking place during remote ischemic conditioning. Activation of transducer and activator of transcription 5 is possibly the end effector, which is responsible for infarct size reduction provided by chronic skeletal muscle ischemia [44].

The morphological study corresponds to the endothelial function markers data. Morphometry showed better-preserved myocardial structure with less infarction, and better-preserved myofibrillar structure in preconditioned animals. Our data confirm the prior study demonstrating the cardioprotective effect of HHP [15]. Data are presented in Figure 7.

In Helmerhorst’s study (2015) the organism’s response to oxygen reduction in the breathing gas mixture from high to normal was called ”pseudohypoxic”, because the term reflects the organism’s universal pathophysiological response to oxygen delivery changes [45]. We studied an analog of pseudohypoxic reaction: first into the hypoxic stage, and then into the hyperoxic stage of preconditioning. Hyperoxic preconditioning leads to an increase of ROS in blood, thereby triggering protective processes in the body. ROS supplements, enhances, and prolongs the effect of hypoxic preconditioning, the protective mechanism of which is implemented along the pseudohypoxic pathway.

In general, tolerance to hyperoxia is species-dependent with smaller animals being lesser resistant to this environmental stress, which may partly explain the varying cardioprotective effects. Hyperoxia alone is an attractive mode of treatment for patients with ischemic heart disease as that breathing hyperoxic gas produces the same results as the ischemic preconditioning without subjecting the patient to transient ischemic conditions, which may cause additive injury.

Presumably, HHP has a combination of modulating action on endothelial function in the condition of IRI on the background of CPB, on the one hand, reducing the ET-1 hyperproduction, on the other hand—stabilizing the formation of nitric oxide metabolites. Thereby, the HHP restoring the balance between vasoconstrictive and vasodilatory properties of endothelial cell products.

However, in many cases, hypoxic and hyperoxic therapies remain clinically marginalized due to the lack of large prospective randomized study outcomes. Recently, we provided a clinical pilot study of the hypoxic-hyperoxic preconditioning (with individual parameters of oxygen selection based on the anaerobic threshold) in patients with coronary artery disease before the main stage of CABG with CPB. Preconditioning allows to reduce the duration of the mechanical ventilation and the catecholamine support and promotes a spontaneous sinus rhythm recovery [46].

Several limitations of the current trial warrant consideration. We are not providing data regarding the optimal duration and titration dose of oxygen treatment, and the duration of exposure. The inclusion of animals with comorbidities could add a confounding variable in the study. We are not providing data regarding the molecular signalling of hypoxic/hyperoxic preconditioning as well as sarcoplasmic reticulum homeostasis. This work represents a pilot study based on the data of endothelial function markers and morphological study in the model of ischemia-reperfusion injury and requires confirmation by molecular and randomized clinical studies to verify the exact mechanism, and the effects of hypoxia and hyperoxia administration in clinical settings involving IRI. The molecular signalling of hypoxic/hyperoxic preconditioning will be the next part of our study.

## 4. Materials and Methods

### 4.1. Design

This is a randomized controlled experimental study of rabbits receiving different types of preconditioning: hypoxic preconditioning (HypP), hyperoxic preconditioning (HyperP), hypoxic-hyperoxic preconditioning (HHP), and control (IRI without preconditioning). Preconditioning was not performed in the control group. The experimental protocol is shown in Figure 8.

### 4.2. Animals and Experimental Procedures

The study was approved by the Institutional Review Board (Project ID code 134, 11/06/2015). This manuscript adheres to the applicable Animal Research: Reporting In Vivo Experiments (ARRIVE) guidelines. All painful procedures were carried out on anesthetized animals according to the Guide for the Care and Use of Laboratory Animals [47]. The study included 32 6-month male rabbits “Soviet Chinchilla”, weighing 3–3.5 kg. The animals were kept under standard vivarium conditions (23 °C, 12 h light/dark cycle) with free access to water and standard chow diet.

The animals were randomly allocated using a closed box containing sealed envelopes, each indicating one of the four groups, just before the baseline data collection. Immediately after randomization, ventilators were adjusted to the assigned settings by an independent biotechnician. Inspiratory oxygen fraction was adjusted and regulated to achieve appropriate concentration throughout the experiment. Researchers were blinded for administered FiO_2_ levels during the experimental procedures. The allocation code of randomization was supplied by the time all data and assay results were collected.

In HypP, we exposed rabbits to two series of 10% oxygen for 10 min with 5 min reoxygenation (21% oxygen). Oxygen was replaced by nitrogen typically to normobaric hypoxic conditions (10% oxygen). In HyperP, the rabbits were exposed to two series of 80% oxygen for 10 min with 5 min reoxygenation (21% oxygen). In HHP, the rabbits were exposed to 10% oxygen for 10 min followed by 80% oxygen for 10 min with 5 min reoxygenation (21% oxygen). In the control group, the rabbits were exposed to 21% oxygen for 25 min. Subsequently, acute myocardial infarction and reperfusion were performed for 45 min and 120 min, respectively, after CPB initiation.

### 4.3. Anaesthesia and Monitoring

All animals were anesthetized by sevoflurane 1.2–1.5 vol% with vaporizer Vapor-2000 (Drager, Germany), intubated and mechanically ventilated through nasotracheal tube №2.5 (original method Patent RU2611955C1). Mechanical ventilation (MV) was carried out using a disposable tube for kids (Intersurgical, UK) with Primus (Drager, Germany), at a tidal volume of 30–40 mL/kg with 5 cm H2O positive end-expiratory pressure, and at a respiratory rate of 50–55 cycles/min. The femoral artery was cannulated with catheter Arteriofix 20G (BBraun, Germany) for continuous blood pressure monitoring and for intermittent arterial blood sampling. The arterial blood pressure was measured with a blood pressure amplifier and data acquisition system (Siemens 7000, Germany) by connecting the catheter to a transducer and calibrating at zero at the mid-chest. The femoral vein was cannulated with Arteriofix 20G catheter (BBraun, Germany) for infusion. Right atrial appendage was used for the venous blood sampling. A temperature sensor was placed into the oesophagus. Moreover, we monitored ECG and pulsoxymetry with a multifunctional monitor (Siemens 7000, Germany).

### 4.4. Samples

All experiments were conducted in the light. Blood analysis (partial tension of oxygen in arterial blood (paO_2_), partial tension of oxygen in venous blood (pvO_2_), saturation of arterial blood (SaO_2_), saturation of venous blood (SvO_2_), oxygen extraction index (IEO_2_), central venous-to-arterial carbon dioxide difference/arterial–venous oxygen content difference ratio (Pcv-aCO_2_/Ca-vO_2_) as an additional indicator of anaerobic metabolism (with cutoff point of 1.8), blood lactate, hemoglobin, and glucose levels was performed on Stat Profile CCX (Nova Biomedical, Waltham, MA, USA) at baseline and every 10 min of the preconditioning procedure.

### 4.5. Cardiopulmonary Bypass Technique

To make the experimental conditions similar to an operating room, we performed the CPB after preconditioning. We used a roll pump with neonatal oxygenator Kids-D100 (Dideco, Italy) in a nonpulsatile mode, the aorta was cannulated by Braunule 16G catheter (BBraun, Germany), right atrial appendage was cannulated by 8F introducer (Edwards Lifesciences, Irvine, CA, USA). The volume velocity of perfusion was calculated with 1.5 L/min/m^2^. Rabbit’s body surface area was 0.25 m^2^. Extracorporeal circulation was performed from aorta to the right atrium. The heart was not unloaded by the pump and functioned at its full capacity. A 100 mL of balanced crystalloid solution (Sterofundin Iso; BBraun Melsungen AG) was used for the initial volume of CPB. Heparin (3 mg/kg) was used before CPB in all cases. Mean arterial blood pressure during CPB was maintained at 40–60 mm Hg due to volume velocity modulation; oesophagus temperature was 36.9 [36.5; 37.1] °C (rabbit’s normal temperature is 38.3–39.5 °C).

### 4.6. Ischemia-Reperfusion Protocol

Acute myocardial infarction and reperfusion were performed for 45 min and 120 min, respectively, after starting CPB without cardioplegia. We induced acute myocardial infarction by ligation of the left coronary artery for 45 min. Ischemia was confirmed by the regional cyanosis downstream of the occlusion by the ST elevation, and by the reduced blood pressure. Reperfusion was confirmed by the lack of cyanosis in that region and by the arrhythmia on ECG.

### 4.7. Endothelial Function Markers

The study of the endothelial function included determination of ET-1 (fmol/mL), asymmetric dimethylarginine (ADMA, µmol/L), endogenous nitrite (NO_2_^−^) and nitrate (NO_3_^−^), and the total concentration of nitric oxide metabolites (NOx total, µmol/L). The study was conducted after the start of CPB within the following phases: (a) baseline, before myocardial ischemia (after preconditioning in experimental groups), (b) after 45 min of myocardial ischemia, and (c) after 120 min of reperfusion. The plasma level of ET-1 was determined by the Biomedica test system (Austria), using solid-phase enzyme-linked immunosorbent assay with the absorption peak of 450 nm. The sensitivity limit of this set is 0.02 fmol/mL (0.05 pg/mL). The plasma level of nitric oxide metabolites (nitrites and nitrates) was determined by the R&D system (R&D Systems Parameter Total NO/Nitrite/Nitrate Kit, Minneapolis, MN, USA) via the enzyme colorimetric method. All samples of one animal were examined with one kit to avoid inter-test variations.

### 4.8. Myocardium Morphology and Morphometry

After 120 min of reperfusion the animals were withdrawn from the experiment with intravenous administration of 10% KCl solution. The left ventricular myocardium containing regions of myocardial infarction, hibernation area, and intact area were harvested and fixed in 10% neutral formalin buffer, followed by paraffin embedding, and preparation of tissue sections (3 μm-thick). Hematoxylin and eosin (H&E) staining were used to observe the changes in myocardial structure. The volume of cardiomyocytes, nuclei, perinuclear vacuolization, interstitial edema, and vessels was measured in the ischemic area and the area at risk of the myocardium by means of an optical microscope (Carl Zeiss Axioskop 40).The hematoxylin basic fuchsin picric acid method was used to detect the early ischemia. The volume of ischemic and non-ischemic cardiomyocytes was measured in the ischemic area and area at risk of the myocardium. The zones were determined visually by changing the color of the myocardium during the stages of ischemia and reperfusion.

Cells were quantified using the National Institutes of Health ImageJ cell counter plugin. At least 12–14 fields of each zone of the heart were quantified. Slides were scanned at ×200 resolution using a digital slide scanner. Multiple microscopic fields of the LV were selected. The visual data of myocardial morphology was subjected to computer morphometry using ImageJ 1.5.

We investigated myocardial slices and measured ischemic area to risk area ratio in our previous study [15]. To determine area of risk (hypoperfusion) ligature tightened again, the heart was stained with 5% solution of potassium permanganate, which is administered through the aortic cannula. In the area of hypoperfusion delimiting zones which are subjected to necrosis of myocardial tissue. The ligature was re-tightened, the heart was stained with a 5% solution of potassium permanganate, which was injected through the aortic cannula (by the modified method of Neckar et al. Basic Res Cardiol. 2002. Vol. 97. p. 161–167). The heart was taken from thoracic cavity, the right ventricle was deleted. The 1mm thick slices were prepared and cut strictly perpendicular to the longitudinal vertical axis of heart, stained with 1% 2,3,5-triphenyltetrazolium chloride (pH 7.4, 37°C) dissolved in 0.1 mol/L phosphate buffer (pH 7.4) for 30 min and fixed overnight in 10 % neutral formaldehyde solution. The slices were scanned (HP Scanjet G4050, Hewlett-Packard, Palo Alto, CA, USA) with 2400 dpi. In the hypoperfusion area, the zones were delineated and the myocardium tissues in these zones were subject to necrosis. The size of ischemic area (IA) and the area at risk (AR) were determined by computerized planimetric method using Ellipse v.2.02 software (ViDiTo, Slovakia).

### 4.9. Statistics

Statistical analysis was carried out in SPSS for Mac v19 (IBM, Inc, USA). Quantitative values were expressed as median and quartiles [25; 75]. A comparison of the quantitative characteristics was performed using the Kruskal–Wallis test with 95% confidence. If the results from the Kruskal-Wallis test were significant, then the Mann–Whitney U-test was similarly applied to analyze each pairing of groups. Intragroup comparisons of the data were carried out using the Wilcoxon test. A *p* value of less than 0.05 was considered statistically significant.

The endothelial markers during the experiment were defined as the primary outcome. Morphological data were assessed as secondary outcomes. Given the lack of published data on this outcome, a sample size calculation was not performed. The animals began to expire after 120 min of reperfusion; therefore, the data acquired until this point after the mechanical ventilation initiation (baseline, after ischemia and after reperfusion) were used for comparisons. At those timepoints, the ET-1, NO_2_, NO_3_^−^, NOx total, and ADMA measurements were all available for analysis. Oxygen balance data was measured and analyzed during the preconditioning procedure. To avoid the unnecessary use of animals, a target *p* value of <0.05 for endothelial markers difference was intended in all interim analyses.

## 5. Conclusions

In conclusion, hyperoxic and hypoxic-hyperoxic preconditioning maintain the endothelial function, balance of nitric oxide metabolites, and reduction of endothelin-1 hyperproduction in the simulation of ischemia-reperfusion myocardial injury in the conditions of cardiopulmonary bypass. The volume of ischemic cardiomyocytes in the HHP group was statistically significantly lesser than that in the control group. Our findings may enhance the understanding of the cardioprotective mechanisms resulting from adaptation to hypoxia and hyperoxia and may lead to the development of new strategies to protect the heart and other organs of patients undergoing cardiac surgery. Further randomized trials are needed to confirm the efficacy of preconditioning treatment and to establish efficacious and safe doses of inspired oxygen fraction and duration of preconditioning procedure in cardiac surgery.

## Figures and Tables

**Figure 1 ijms-21-05336-f001:**
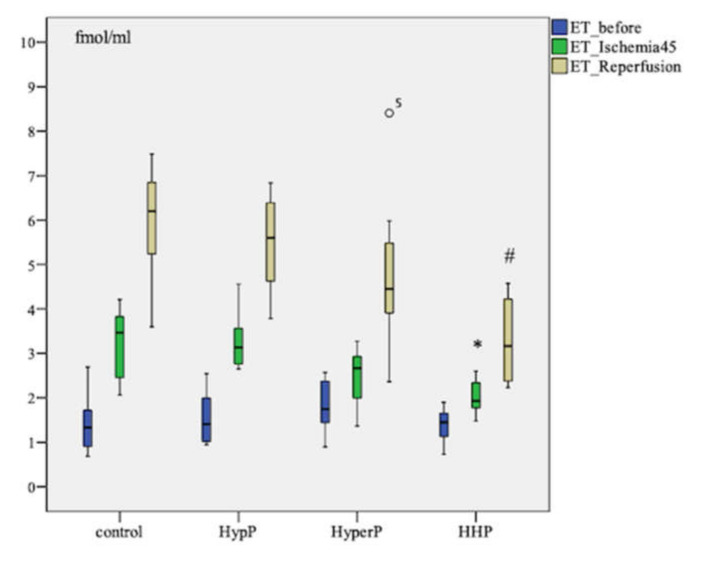
Dynamics of endothelin-1 (ET-1) concentration in blood plasma in animals during the experiment with different types of preconditioning, fmol/mL (*n* = 32, *n* = 8 rabbits/group). HypP—hypoxic preconditioning; HyperP—hyperoxic preconditioning; HHP—hypoxic-hyperoxic preconditioning; control (without preconditioning). Data are expressed as median and quartiles [25; 75]; the circle number 5 is missed value; * *p* = 0.006 HHP vs control group; ^#^
*p* = 0.003 HHP vs control group; the Mann-Witney test.

**Figure 2 ijms-21-05336-f002:**
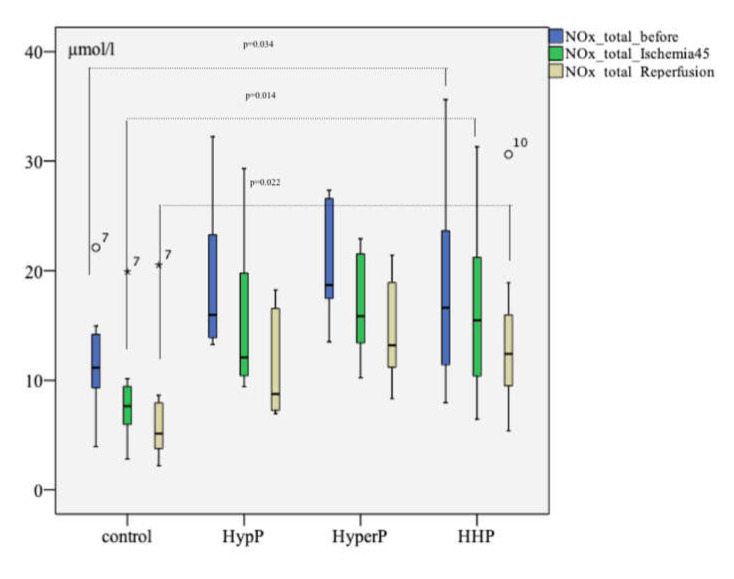
The total concentration of nitric oxide metabolites (NOx.total) during the experiment with different types of preconditioning, µmol/L (*n* = 32, *n* = 8 rabbits/group). The total concentration of nitric oxide metabolites before the ischemia but after different types of preconditioning was higher than that in control group, as well as after 45 min of ischemia and 120 min of reperfusion; the circles number 7 and 10, and asterisk number 7 are missed values; *p* was pointed on between the HHP and control groups analysis at the study stages; the Mann-Witney test.

**Figure 3 ijms-21-05336-f003:**
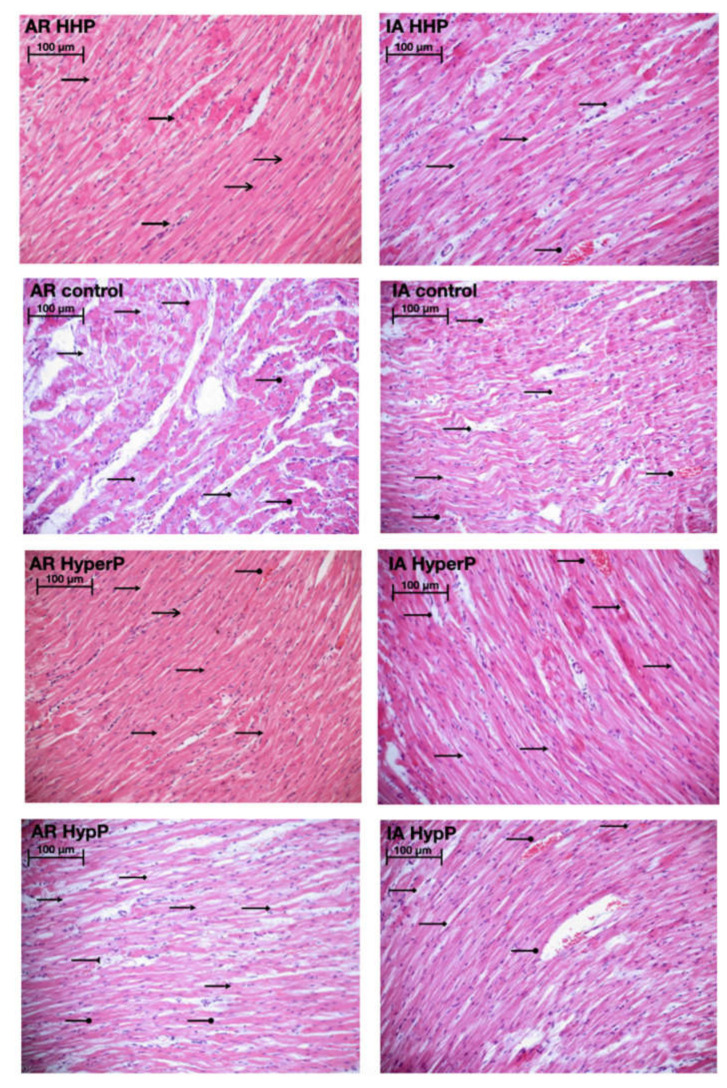
Light microscopy of the myocardium of the left ventricle revealed disturbances of myocardial structure, increase in the distance between the discs, and a lack of cross-striations in the hibernation area. We observed less intense damage in HHP animals compared with the control group. Area at risk (AR) HHP group: slight dystrophy of cardiomyocytes, obvious cross-striation (
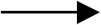
), preserved myofibrils; only a few pyknotic nuclei (
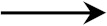
); Ischemic area (IA) HHP group: hyperaemia, interstitial edema (
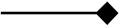
), perinuclear vacuolization (
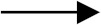
), enlarged capillaries (
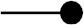
);AR Control group: severe dystrophy (
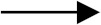
) of cardiomyocytes, loss of cross-striation (
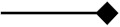
), partial degradation of myofibrils; massive cariopicnose (
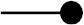
); IA Control group: hyperaemia, interstitial edema (
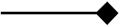
), perinuclear vacuolization (
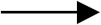
), partial degradation of myofibrils, enlarged capillaries (
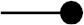
), local haemorrhages; AR HyperP group: dystrophy of cardiomyocytes, preserved cross-striation and myofibrils (
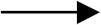
); a few pyknotic nuclei (
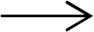
), enlarged capillaries (
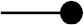
); IA HyperP group: hyperaemia, interstitial edema (
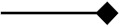
), perinuclear vacuolization (
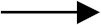
), enlarged capillaries (
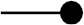
), dystrophy (
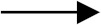
) of cardiomyocytes; AR HypP group: dystrophy of cardiomyocytes (
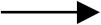
), loss of cross-striation (
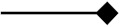
), partial degradation of myofibrils; cariopicnose (
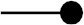
), interstitial edema (
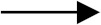
); IA HypP group: hyperaemia, interstitial edema (
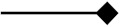
), partial degradation of myofibrils, enlarged capillaries (
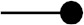
), loss of cross-striation (
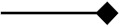
), local haemorrhages; 200×, H&E.

**Figure 4 ijms-21-05336-f004:**
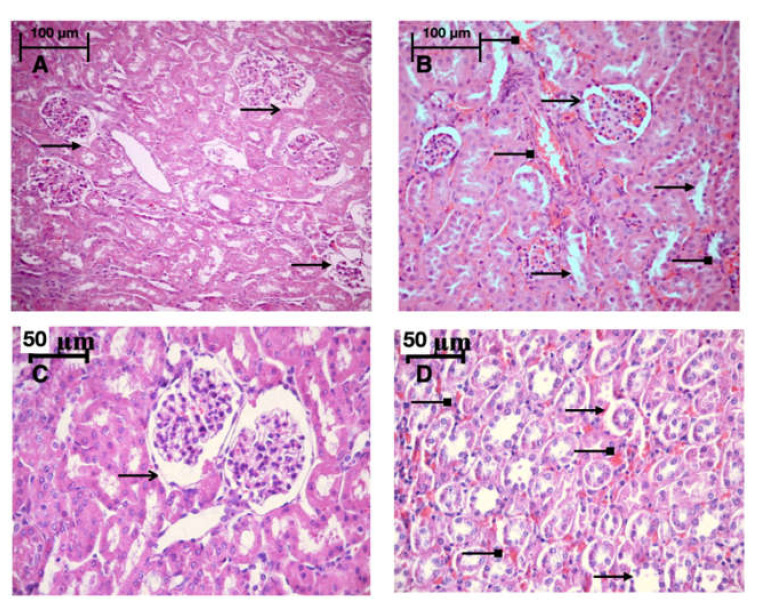
Light microscopy of the kidneys, they were less affected in the HHP group then in the control group. HHP group (**A**): a moderate edema (
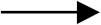
), the capsule of the glomerulus (
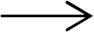
) is moderately expanded, 200×, H&E; Control group (**B**): marked edema (
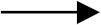
), enlarged capillaries (
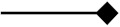
) of cortical and medullar substances, the capsule of the glomerulus (
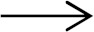
) is moderately expanded, 200×, H&E; HyperP group (**C**): the nuclei of the podocytes are normal, the capsule of the glomerulus (
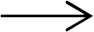
) is moderately expanded, a moderate amount of red blood cells in the capillaries. 400×, H&E; HypP group (**D**): enlarged capillaries (
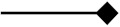
) of cortical substance, edema (
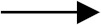
), destructive changes in cells. 400×, H&E.

**Figure 5 ijms-21-05336-f005:**
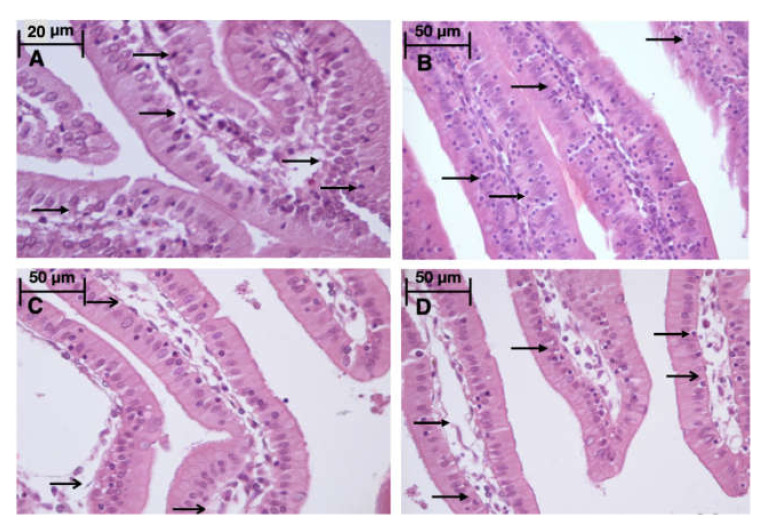
Light microscopy of the gut mucosa. HHP group (**A**): a large number of intraepithelial lymphocytes (
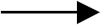
), 670x, H&E; Control group (**B**): a large number of intraepithelial lymphocytes (
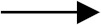
), 400×, H&E; HyperP group (**C**): intestinal villus edema (
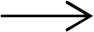
), 400×, H&E; HypP group (**D**): intestinal villi edema (
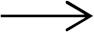
), a large number of intraepithelial lymphocytes (
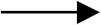
), 400×, H&E.

**Figure 6 ijms-21-05336-f006:**
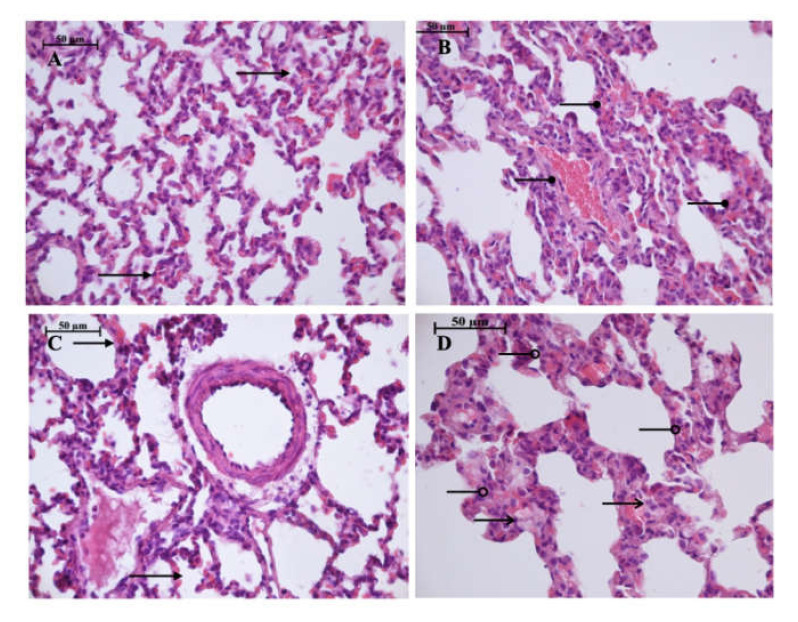
Light microscopy of the lung parenchyma. HHP group (**A**): moderate hyperemia (
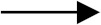
) of the interalveolar capillaries, 400×, H&E; Control group (**B**): enlarged capillaries (
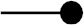
), filled with erythrocytes, 400×, H&E; HyperP group (**C**): moderate hyperemia (
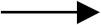
) of the interalveolar capillaries, 400×, H&E; HypP group (**D**): edema (
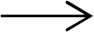
) of the interalveolar septa, dystrophy of alveoles, atelectasis (
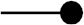
), 400×, H&E.

**Figure 7 ijms-21-05336-f007:**
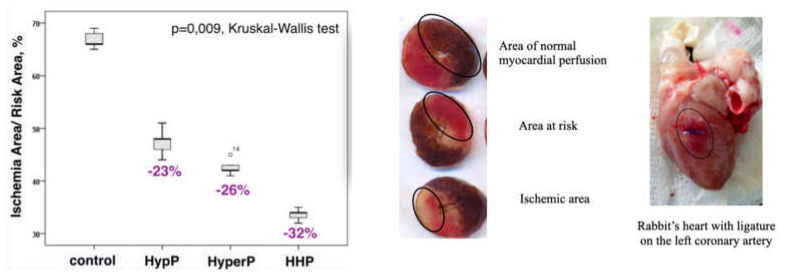
Infarct size was reduced significantly in all preconditioning groups compared with the control group. HypP—hypoxic preconditioning; HyperP—hyperoxic preconditioning; HHP—hypoxic-hyperoxic preconditioning; control (without preconditioning). The ischemic area to area at risk ratio (IA/RA) decreased by 23% in the HypP group, by 26% in the HyperP group, by 32% in the HHP group vs control group (*p* = 0.009).

**Figure 8 ijms-21-05336-f008:**
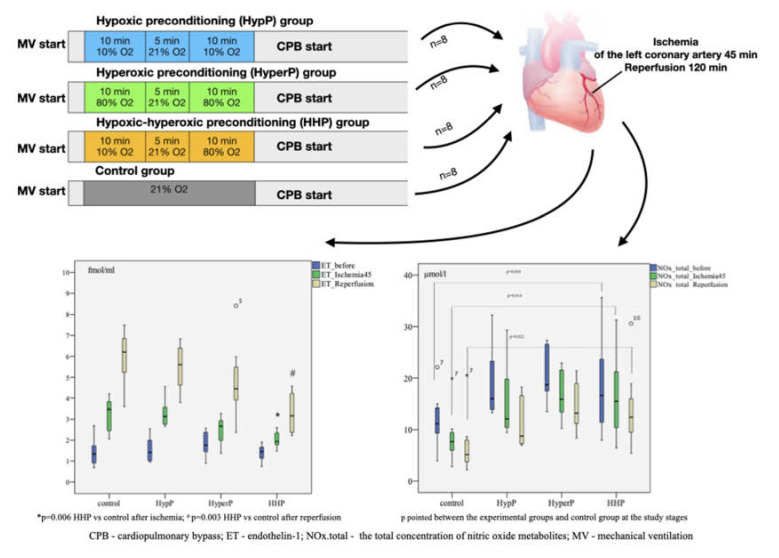
A randomized controlled experimental study of rabbits (*n* = 32) receiving different types of preconditioning: hypoxic preconditioning (HypP), hyperoxic preconditioning (HyperP), hypoxic-hyperoxic preconditioning (HHP), and control (Ischemia-Reperfusion Injury without preconditioning). Subsequently, acute myocardial infarction and reperfusion were performed for 45 min and 120 min, respectively, after CPB initiation. The hyperoxic and hypoxic-hyperoxic preconditioning maintain the endothelial function, balance of nitric oxide metabolites and reduction of endothelin-1 hyperproduction.

**Table 1 ijms-21-05336-t001:** Baseline oxygen balance characteristics before the preconditioning procedure (*n* = 32).

Data	Hypoxic Preconditioning, *n* = 8	Hyperoxic Preconditioning, *n* = 8	Hypoxic-hyperoxic Preconditioning, *n* = 8	Control, *n* = 8	*p*
paO_2__, mmHg_	116 [108.8; 187]	187 [121.0; 196.8]	142 [125.8; 175.8]	187 [114; 196.8]	0.455
pvO_2__, mmHg_	42 [39.0; 48.8]	42 [39.5; 44.3]	39 [38.3; 48.8]	40.5 [39.0; 41.8]	0.760
dPv-aCO_2_ /Ca-vO_2_	0.8 [0.7; 0.9]	1.1 [0.9; 1.2]	0.9 [0.8; 1.1]	0.9 [0.8; 1.1]	0.249
SaO_2_	98 [96.0; 98.8]	98.5 [97.3; 99.0]	97 [96.0; 98.0]	98.5 [97.3; 99.8]	0.194
SvO_2_	54.5 [48; 72]	59 [55.5; 65.0]	55 [49.0; 68.3]	56 [48.0; 60.8]	0.597
IEO_2_	43.8 [26.7; 50.4]	40.4 [34.5; 44.1]	43.3 [30.2; 50.2]	43.1 [37.3; 51.4]	0.704
Glu, mmol/L	7.5 [6.0; 8.0]	6 [5.0; 6.8]	7 [6.0; 7.8]	5.5 [5.0; 6.8]	0.075
Lac, mmol/L	4 [3.1; 5.0]	3.5 [3.0; 5.0]	3.2 [3.0; 4.6]	4.5 [3.3; 5.0]	0.533

Comments: MAP—mean arterial pressure; paO2—partial tension of oxygen in arterial blood; pvv2—partial tension of oxygen in venous blood; dPv-aCO2/Ca-vO2—delta partial tension of central venous-to-arterial carbon dioxide difference/arterial–venous oxygen content difference ratio; SaO2—saturation of arterial blood; SvO2—saturation of venous blood; IEO2—oxygen extraction index; Glu—glucose level in plasma; Lac—lactate level in plasma. Values are shown as median and [25; 75 quartile]. *p* was pointed on between group comparison, the Kruskal-Wallis test.

**Table 2 ijms-21-05336-t002:** Physiological and Hemodynamic variables (*n* = 32).

Data	Hypoxic Preconditioning, *n* = 8	Hyperoxic Preconditioning, *n* = 8	Hypoxic-Hyperoxic Preconditioning, *n* = 8	Control, *n* = 8	*p*
**Physiological variable**					
Oesophagus temperature, °C	36.5 [36.4; 37.0]	37 [36.7; 37.2]	36.9 [36.6; 37.0]	37.1 [36.6; 37.4]	0.181
	**Hemodynamic variables**				
Heart rate, beats/min	180.5 [167.8; 187.8]	179 [176.5; 186.3]	191 [186.5; 202.0]	190.5 [171.8; 193.0]	0.131
Mean arterial pressure, mmHg	69.5 [68.3; 72.5]	65.5 [60.0; 70.3]	70 [62.5; 73.3]	68 [65.0; 71.5]	0.474
Hemoglobin, g/L	91.5 [89.3; 109]	100.5 [98.3; 108.3]	96 [90.5; 102.5]	97.5 [91.8; 103.8]	0.517
Diuresis, mL/kg/h	1.6 [1.5; 1.7]	1.35 [1.05; 1.63]	1.3 [1.02; 1.68]	1.45 [1.23; 1.58]	0.195

Values are shown as median and [25; 75 quartile]. *p* was pointed on between group comparison, the Kruskal-Wallis test.

**Table 3 ijms-21-05336-t003:** Oxygen balance characteristics during the preconditioning procedure.

Data	Hypoxic Preconditioning, *n* = 8	Hyperoxic Preconditioning, *n* = 8	Hypoxic-hyperoxic Preconditioning, *n* = 8	
	20 min	20 min	Hypoxic Phase, 10 min	Hyperoxic Phase, 10 min
paO_2__, mmHg_	50.5 [49; 52.8] *	348 [317.5; 392.5] *	52 [48.3; 58.8] *	340 [310; 375] *
pvv_2__, mmHg_	39 [37.3; 40]	48 [46; 50.8]	37 [37; 39.5]	47.5 [44.3; 52]
dPv-aCO_2_ /Ca-vO_2_	0.8 [0.7; 0.9]	1.1 [0.9; 1.2]	1.0 [0.8; 1.1]	1.1 [0.9; 1.2]
SaO_2_	82.5 [77; 85]*	99 [99; 100]	82 [77.3; 85.5]*	99 [99; 99]
SvO_2_	50 [49; 52.3]	64.5 [56.3; 68.8]	50 [48.3; 53.3]	59 [56.5; 67]
IEO_2_	38.9 [33.4; 42.1]	35.2 [31.3; 43.2]	38.9 [33.4; 42.3]	40.4 [31.8; 42.9]
Glu, mmol/L	6 [5.3; 6.8]	6 [4; 7]	4.5 [4; 6.8]	7 [5.3; 8]
Lac, mmol/L	3 [3; 3.8]	5 [4.3; 6]	4 [3; 4.8]	5 [4; 6]

Comments: paO2—partial tension of oxygen in arterial blood; pvv2—partial tension of oxygen in venous blood; dPv-aCO2/Ca-vO2—delta partial tension of central venous-to-arterial carbon dioxide difference/arterial–venous oxygen content difference ratio; SaO2—saturation of arterial blood; SvO2—saturation of venous blood; IEO2—oxygen extraction index; Glu—glucose level in plasma; Lac—lactate level in plasma. Values are shown as median and [25; 75 quartile]. Intra group comparisons of the data were carried out using the Wilcoxon test, *—*p* < 0,05 comparing to the baseline.

**Table 4 ijms-21-05336-t004:** Characteristics of nitric oxide metabolites and ADMA concentration at the study stages.

Parameter	HypP, *n* = 8	HyperP, *n* = 8	HHP, *n* = 8	Control, *n* = 8	*p* # between Groups
Changes in nitrite concentration					
NO_2_.endo, before ischemia, µmol/L	0.947 [0.806; 1.187]	0.822 [0.585; 0.976]	1.095 [0.906; 1.331]	0.442 [0.366; 0.573]	0.010
NO_2_.endo, after ischemia, µmol/L	0.732 [0.572; 0.857]	0.627 [0.555; 0.811]	0.882 [0.631; 1.042]	0.266 [0.217; 0.345]	0.005
*p* * in comparison with the initial value	*p* = 0.012	*p* = 0.069	*p* = 0.012	*p* = 0.012	
NO_2_.endo, after reperfusin, µmol/L	0.511 [0.378; 0.640]	0.505 [0.409; 0.633]	0.706 [0.590; 0.838]	0.203 [0.180; 0.319]	0.002
*p* * in comparison with the initial value	*p* = 0.012	*p* = 0.161	*p* = 0.012	*p* = 0.012	
Changes in nitrate concentration					
NO_3_.endo, before ischemia, µmol/L	14.803 [12.756; 22.903]	18.033 [16.247; 25.839]	13.808 [10.223; 24.836]	10.732 [8.740; 14.317]	0.045
NO_3_.endo, after ischemia, µmol/L	11.451 [9.595; 19.366]	15.214 [12.764; 21.260]	13.995 [9.019; 22.961]	7.399 [5.570; 9.664]	0.027
*p* * in comparison with the initial value	*p* = 0.012	*p* = 0.012	*p* = 0.484	*p* = 0.012	
NO_3_.endo, after reperfusion, µmol/L	8.177 [6.727; 16.170]	12.202 [10.187; 18.981]	10.832 [8.683; 16.600]	4.938 [3.526; 8.168]	0.026
*p* * in comparison with the initial value	*p* = 0.012	*p* = 0.012	*p* = 0.069	*p* = 0.012	
Changes in ADMA concentration					
ADMA, before ischemia, µmol/L	1.659 [1.400; 2.593]	1.692 [1.567; 1.997]	1.617 [1.449; 1.944]	1.746 [1.255; 2.147]	0.827
ADMA, after ischemia, µmol/L	1.62 [1.50; 2.395]	1.781 [1.564; 1.985]	1.684 [1.328; 1.893]	2.310 [2.092; 2.732]	0.036
*p* * in comparison with the initial value	*p* = 0.575	*p* = 0.575	*p* = 0.484	*p* = 0.036	
ADMA, after reperfusion, µmol/L	1.727 [1.680; 2.322]	1.79 [1.571; 1.890]	1.803[1.450; 1.956]	2.325 [1.864; 2.678]	0.084
*p* * in comparison with the initial value	*p* = 0.779	*p* = 0.263	*p* = 0.674	*p* = 0.050	

HypP—hypoxic preconditioning; HyperP—hyperoxic preconditioning; HHP—hypoxic-hyperoxic preconditioning; control (without preconditioning); NO2.endo—endogenous nitrite (NO2); NO3.endo—endogenous nitrate (NO3); ADMA—asymmetric dimethylarginine. Values are shown as median and [25; 75 quartile]. *—*p* was pointed on intragroup analysis compared to the initial data (before ischemia), the Wilcoxon test; #—*p* was pointed on between group comparison, the Kruskal-Wallis test.

**Table 5 ijms-21-05336-t005:** The volume of parenchymal and stromal components of cardiomyocytes (μm^3^/μm^3^) in Hypoxic-hyperoxic preconditioning and control groups.

	Ischemic Area, Control Group, *n* = 8	Ischemic Area, HHP Group, *n* = 8	Area at Risk, Control Group, *n* = 8	Area at Risk, HHP Group, *n* = 8
Cardiomyocyte, μm^3^/μm^3^	0.687 [0.632; 0.742]	0.722 [0.643; 0.758]	0.603 [0.550; 0.694]	0.718 [0.679; 0.782] *
Nuclei, μm^3^/μm^3^	0.057 [0.029; 0.083]	0.046 [0.027; 0.080]	0.035 [0.027; 0.045]	0.052 [0.024; 0.068]
Perinuclear vacuolization, μm^3^/μm^3^	0.047 [0.028; 0.068]	0.027 [0.011; 0.054] *	0.046[0.019; 0.071]	0.025 [0.010; 0.050] *
Interstitial oedema, μm^3^/μm^3^	0.128 [0.083; 0.168]	0.102 [0.067; 0.152]	0.17 [0.135; 0.232]	0.091 [0.062; 0.129] *
Vessels,μm^3^/μm^3^	0.057 [0.025; 0.101]	0.076 [0.035; 0.103]	0.114 [0.093; 0.145]	0.067 [0.035; 0.103] *

*—*p* < 0.05 between groups. Values are shown as median and [25; 75 quartile].

**Table 6 ijms-21-05336-t006:** The volume of ischemic and non-ischemic cardiomyocytes (μm3/μm^3^) in Hypoxic-hyperoxic preconditioning and control groups.

Cardiomyocytes	Ischemic Area, Control Group, *n* = 8	Ischemic Area, HHP Group, *n* = 8	Area at Risk, Control Group, *n* = 8	Area at Risk, HHP Group, *n* = 8
Ischemic, μm^3^/μm^3^	0.432 [0.313; 0.533]	0.329 [0.240; 0.438] *	0.481 [0.346; 0.577]	0.268 [0.112; 0.361] *
Non-ischemic, μm^3^/μm^3^	0.568 [0.467; 0.687]	0.671 [0.562; 0.760] *	0.519 [0.423; 0.654]	0.732 [0.639; 0.888] *

*—*p* < 0.05 between groups. Values are shown as median and [25; 75 quartile].

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
