# Peer review of "Influence of Hypoxic and Hyperoxic Preconditioning on Endothelial Function in a Model of Myocardial Ischemia-Reperfusion Injury with Cardiopulmonary Bypass (Experimental Study)"

_ijms, 2020, doi:10.3390/ijms21155336_

Round 1
Reviewer 1 Report
Authors have made an interesting effort investigating the regulation of Endothelin-1 and the NO homeostasis during ischemia-reperfusion in terms of a clinically applicable hypoxic/hyperoxic ischemic preconditioning. However, there are major issues concerning the manuscript layout as many of the figures and figure legends have been relocated making the manuscript illegible to many parts. Moreover, some scientific issues emerge.
Major Comments
- Results presented in the current manuscript is mostly descriptive. Authors have determined the levels of ET-1, NO by-products and ADMA in the plasma of the experimental animals. However, the direct effects of the hypoxic/hyperoxic preconditioning on the myocardium is only investigated through histology. It would be interesting if authors could address the molecular signalling of hypoxic/hyperoxic preconditioning, in terms of eNOS phosphorylation status or it’s upstream effectors Akt and PI3K (Bibli SI et al. J Cardiovasc Pharmacol Ther. 2014 Mar;19(2):220-7). This would facilitate the mechanistic insight between NO release and myocardial damage.
- ET-1 has been shown to disturb Ca2+ homeostasis and induce inotropy and Ca2+-mediated apoptosis in the cardiomyocytes (Vignon-Zellweger N, et al. Life Sci. 2012). It would be therefore interesting if authors could -alongside eNOS signalling- investigate sarcoplasmic reticulum homeostasis through studying key SR molecules regulation, such as SERCA, Ryanodine Receptor and phospholamban .
- The most concerning issue of the manuscript is the translational value of the protocol. Even though preconditioning is proven to be a highly cardioprotective mechanism, its translational value is minimal as in clinical praxis, it is highly impossible to foresee an upcoming acute myocardial infarction and apply a prompt intervention. It would be interesting if authors could investigate the same cardioprotective potential in terms of ischemic postconditioning, which is more clinically relevant.
Minor Comments
- Authors should address the grammatical errors within the text such as “disregulation” to “dysregulation p.2, l. 77 and “decrease” to “decreases” p.2, l. 91. It would be preferable if a native English speaker could revise the linguistic deficits of the manuscript.
- The figures and figure legends have been moved within the text and therefore, figure legends are currently illegible. Authors should carefully revise the manuscript layout upon pdf merging.
- The methodology for infarct size estimation is not clearly presented. Judging for the images in the discussion section authors have performed TTC and maybe Evan’s blue staining. The details of the staining should be described.
- In the tables of the manuscript, authors are advised to provide the n numbers of the groups beside the group names, in order to make the results more descriptive and transparent.
Author Response
Response to Reviewer 1 Comments
Major Comments
Point 1: Results presented in the current manuscript is mostly descriptive. Authors have determined the levels of ET-1, NO by-products and ADMA in the plasma of the experimental animals. However, the direct effects of the hypoxic/hyperoxic preconditioning on the myocardium is only investigated through histology. It would be interesting if authors could address the molecular signalling of hypoxic/hyperoxic preconditioning, in terms of eNOS phosphorylation status or it’s upstream effectors Akt and PI3K (Bibli SI et al. J Cardiovasc Pharmacol Ther. 2014 Mar;19(2):220-7). This would facilitate the mechanistic insight between NO release and myocardial damage.
Response 1: Thank you for your valuable comments and support of our manuscript! Certainly it would be interesting to address the molecular signalling of preconditioning and to discover the exact mechanism of preconditioning effect.
Bibli SI et al. study revealed the cardioprotective molecular signalling of chronic remote (hind limb) ischemia combined or not with ischemic pre- and postconditioning in a model of ischemic-reperfusion injury in rabbits. There is an interesting idea: «In clinical practice, chronic «hind limb» ischemia cannot be applied to patients before AMI. However, there are many patients with coronary artery disease who have severe peripheral atherosclerosis leading to sequential subtotal lesions or even total occlusions of the iliac, femoral, tibial, or other arteries. Another group of patients are those who have been subjected to limb amputation for various reasons. According to our findings, such patients are already in a ‘‘conditioning state’’ and therefore more tolerant against myocardial ischemia/reperfusion regardless of the application of coronary conditioning». This study is similar to ours. We all trying to provide experimental methods of organ protection aiming to help people to survive through critical illness. In our study we investigate the methods of organ protection during cardiosurgery with cardiopulmonary bypass. Cardiopulmonary bypass is an inalienable part of the operation during which the ischemia-reperfusion injury occurs inevitably.
The molecular signalling of hypoxic/hyperoxic preconditioning will be the next part of our study. Unfortunately, we have no opportunity to do such investigation on this material. The morphology, as basic method of study of tissue structure, shows appropriate difference between preconditioning methods.
Point 2: ET-1 has been shown to disturb Ca2+ homeostasis and induce inotropy and Ca2+-mediated apoptosis in the cardiomyocytes (Vignon-Zellweger N, et al. Life Sci. 2012). It would be therefore interesting if authors could -alongside eNOS signalling- investigate sarcoplasmic reticulum homeostasis through studying key SR molecules regulation, such as SERCA, Ryanodine Receptor and phospholamban.
Response 2: Undoubtedly it would be interesting to investigate sarcoplasmic reticulum homeostasis through studying key SR molecules regulation. In Vignon-Zellweger N, et al. review of Endothelin and endothelin receptors in the renal and cardiovascular systems clearly described the mechanisms of vasoconstriction and vasodilatation. «The ET-1 is a multifunctional hormone with complex effects on the renal, cardiac and vascular physiology. ETA and ETB receptors can have synergetic or opposing effects depending on cell type, tissue type or physiological situation. Both ETA and ETB receptors are expressed throughout the cardiac muscle, including the coronary vasculature. Cardiomyocytes express predominantly ETA receptors. ET-1 is a strong vasoconstrictor while the activation of the endothelial ETB receptor induces the production of vasodilator NO. Activation of the ETA receptor results in increased contractility. ET-1 is also responsible for the positive inotropic effect of angiotensin II via the production of reactive oxygen species. ET-1 is overexpressed in the failing heart but may prevent apoptosis and restore cardiac function in stress situation».
In our study we observed the vasoconstriction and vasodilatation effects of changes in ET-1 and NO production during stress (ischemia-reperfusion injury) on the background of preconditioning or without it. So we hypothesise that changes in vascular tone are the basis of vascular training before stress situation. We are looking forward to provide more studies (including patients) to establish the exact vascular reaction and molecular signalling of preconditioning in clinical practice.
Point 3: The most concerning issue of the manuscript is the translational value of the protocol. Even though preconditioning is proven to be a highly cardioprotective mechanism, its translational value is minimal as in clinical praxis, it is highly impossible to foresee an upcoming acute myocardial infarction and apply a prompt intervention. It would be interesting if authors could investigate the same cardioprotective potential in terms of ischemic postconditioning, which is more clinically relevant.
Response 3: Of course it is impossible to foresee an upcoming acute myocardial infarction. The postconditioning is more clinically relevant for this patients. But the different preconditioning methods are proven in many studies to be a highly cardioprotective. Its translational value is clinically relevant especially in cardiosurgery. Cardiosurgery patients are often old and have comorbidities. So, the cardio operation is hazardous to the length and quality of life of the patients, because of ischemia-reperfusion injury that is inevitably during cardiopulmonary bypass (CPB). The duration of CPB is depends on the type of the operation and the ability of the surgeon to do all the best as fast as possible. Many operations lasts more then 5 hours with CPB of 60-120 minutes. The systemic inflammatory reaction syndrome is manifested during nonpulsatile mode of CPB. Not only myocardium but other organs are affected with hypoperfusion and inflammation during CBP. The preconditioning methods are able to improve or prevent the organs damage during CPB.
Minor Comments
Point 1: Authors should address the grammatical errors within the text such as “disregulation” to “dysregulation p.2, l. 77 and “decrease” to “decreases” p.2, l. 91. It would be preferable if a native English speaker could revise the linguistic deficits of the manuscript.
Response 1: The native English speaker revised the manuscript.
Point 2: The figures and figure legends have been moved within the text and therefore, figure legends are currently illegible. Authors should carefully revise the manuscript layout upon pdf merging.
Response 2: We are so sorry for the mass with figures and legends. We have revised the manuscript layout and fixed it.
Point 3: The methodology for infarct size estimation is not clearly presented. Judging for the images in the discussion section authors have performed TTC and maybe Evan’s blue staining. The details of the staining should be described.
Response 3: You are absolutely right. We didn’t describe this method. We investigated myocardial slices and measured ischemic area to risk area ratio in our previous study. We introduced the corresponding changes to the manuscript.
We investigated myocardial slices and measured ischemic area to risk area ratio in our previous study [12]. To determine area of risk (hypoperfusion) ligature tightened again, the heart was stained with 5% solution of potassium permanganate, which is administered through the aortic cannula. In the area of hypoperfusion delimiting zones which are subjected to necrosis of myocardial tissue. The ligature was re-tightened, the heart was stained with a 5% solution of potassium permanganate, which was injected through the aortic cannula (by the modified method of Neckar et al.// Basic Res Cardiol. 2002. Vol. 97. р. 161 – 167). The heart was taken from thoracic cavity, the right ventricle was deleted. The 1mm thick slices were prepared and cut strictly perpendicular to the longitudinal vertical axis of heart, stained with 1% 2,3,5-triphenyltetrazolium chloride (pH 7.4, 37°C) dissolved in 0.1 mol/l phosphate buffer (pH 7.4) for 30 min and fixed overnight in 10 % neutral formaldehyde solution. The slices were scanned (scanner «HP Scanjet G4050 (Hewlett-Packard, Palo Alto, USA) with 2400 dpi. In the hypoperfusion area, the zones were delineated and the myocardium tissues in these zones were subject to necrosis. The size of ischemic area (IA) and the area at risk (AR) were determined by computerized planimetric method using Ellipse v.2.02 software (ViDiTo, Slovakia).
Point 4: In the tables of the manuscript, authors are advised to provide the n numbers of the groups beside the group names, in order to make the results more descriptive and transparent.
Response 4: Thank you for advise to provide the n numbers of the groups beside the group names. We introduced the corresponding changes to the Tables.

Reviewer 2 Report
The aim of the study was to estimate the effect of preconditioning based on changes in inspiratory oxygen fraction on endothelial function in the model of IRI of the myocardium under conditions of cardiopulmonary bypass (CPB). The primary finding is that preconditioning with hypoxia, hyperoxia and/or their combination modulates endothelial function. The authors propose that the preconditioning with different fraction of inspired oxygen protects the function of the endothelium by way of reduction of ET-1 hyperproduction in response to stress including IRI. Second, preconditioning was shown to enhance total concentration of nitric oxide metabolites in animals after IRI indicating protective properties through maintaining the concentration of NO metabolites.
The study contributes to the understanding of some potential mechanisms underlying the mechanisms of preconditioning. However, I have some comments and questions, since some formulations are not clear.
General comments
It would be helpful to show the experimental protocol in a graphical form.
It is no clear why all the values are presented as median and quartiles. It makes it sometimes difficult to understand the differences between the groups.
Specific comments
1). INTRODUCTION – is too long
2). REFERENCES - please add 2 more Clinical refs that can support the idea of cardiopulmonary bypass prior to acute myocardial infarction. In clinical situation, CPB is started after the occurrence of AMI. What was the idea to start CPB and then perform the AMI? This is different from the clinical setting.
3). What are findings in female population? Why are the authors choosing male only for this study?
4) It would be helpful to show all hemodynamic and biometric parameters of animals at the baseline pre-ischemic conditions in a separate Table.
4). Tables 1&2; Figures 1 & 2 & 5 are all showing results for 4 investigated groups HOWEVER the light micrographs shown in Figures 3&4 and results in Tables 3&4 show only two groups (Control & HHP).
WHAT WERE THE RESULTS LIKE FOR GROUPS HypP & HyperP?
5). IN FIGURE 3, in all images (ABCD) it would be beneficial to insert an arrow to point out important difference between the Control and HHP.
6). Scale is needed for all light micrographs (photos).
7). In Fig 3 if there is a slight dystrophy of cardiomyocytes - did the authors look into the signs of inflammation and fibrosis? (That’s what I have found in literature now).
8) How was the size of infarction evaluated without well-known method of TTC staining?
9). IN CONCLUSION - the results are not clearly presented and major revision is recommended.
Author Response
General comments
Point 1: It would be helpful to show the experimental protocol in a graphical form.
Response 1: Thank you for your valuable comments and support of our manuscript! We introduced the corresponding changes to the manuscript and added Figure 8.
The native English speaker revised the manuscript.
We are so sorry for the mass with figures and legends. We have revised the manuscript layout and fixed it.
Point 2: It is no clear why all the values are presented as median and quartiles. It makes it sometimes difficult to understand the differences between the groups.
Response 2: We investigated a small groups of animals (n=8 in each group), because of that the data did not meat the normal distribution criteria, the values were presented as median and quartiles.
Specific comments
Point 1: INTRODUCTION – is too long
Response 1: We shortened Introduction section as required.
Point 2: REFERENCES - please add 2 more Clinical refs that can support the idea of cardiopulmonary bypass prior to acute myocardial infarction. In clinical situation, CPB is started after the occurrence of AMI. What was the idea to start CPB and then perform the AMI? This is different from the clinical setting.
Response 2: We added 3 Clinical refs. The idea to start CPB and then perform the AMI is similar to an operation stages. We performed the preconditioning, then we started CPB. In clinical circumstances the main stage of the cardiac operation provides after aorta cross-clamping and a cardioplegia solution infusion. The quality of cardioprotection with different types of cardioplegia solutions depends on many factors (coronary atherosclerosis, heart anatomy, a way of cardioplegia solution delivery, a length of the aorta cross-clamping). Intraoperative myocardial dysfunction occurs up to 25%, the postoperative AMI occurs up to 8%. The ischemia-reperfusion injury is inevitably during CPB. The duration of CPB is depends on the type of the operation and the ability of the surgeon to do all the best as fast as possible. Many operations lasts more then 5 hours with CPB of 60-120 minutes. The systemic inflammatory reaction syndrome is manifested during nonpulsatile mode of CPB. Not only myocardium but other organs are affected with hypoperfusion and inflammation during CBP. The preconditioning methods are able to improve or prevent the organs damage during CPB.
Point 3: What are findings in female population? Why are the authors choosing male only for this study?
Response 3: Thank you for your comment. It would be very interesting to compare male and female. We choose male only just to make the group more homogeneous.
Point 4: It would be helpful to show all hemodynamic and biometric parameters of animals at the baseline pre-ischemic conditions in a separate Table.
Response 4: We introduced the corresponding changes to the manuscript and added revised Tables 1-3.
Point 5: Tables 1&2; Figures 1 & 2 & 5 are all showing results for 4 investigated groups HOWEVER the light micrographs shown in Figures 3&4 and results in Tables 3&4 show only two groups (Control & HHP).
WHAT WERE THE RESULTS LIKE FOR GROUPS HypP & HyperP?
Response 5: The Figures 3&4 have been changed according to the reviewer’s comment. We introduced the corresponding changes to the manuscript and added Figures 3-6. In Tables 3&4 we presented data from two groups (HHP and Control), as the biggest differences were revealed in them (in order to make the data easy to understand).
Point 6: IN FIGURE 3, in all images (ABCD) it would be beneficial to insert an arrow to point out important difference between the Control and HHP.
Response 6: The Figures have been changed according to the reviewer’s comment.
Point 7: Scale is needed for all light micrographs (photos).
Response 7: The Figures have been changed according to the reviewer’s comment.
Point 8: In Fig 3 if there is a slight dystrophy of cardiomyocytes - did the authors look into the signs of inflammation and fibrosis? (That’s what I have found in literature now).
Response 8: We found the hyperaemia, interstitial edema, and perinuclear vacuolization in the ischemic area of the HHP group of animals. We found the enlarged capillaries, local hemorrhages, interstitial edema, and perinuclear vacuolization in the ischemic area of the control group of animals. There were no signs of fibrosis.
Point 9: How was the size of infarction evaluated without well-known method of TTC staining?
Response 9: The zones of myocardial infarction, hibernation area, and intact area were determined visually by changing the color of the myocardium during the stages of ischemia and reperfusion.
Point 10: IN CONCLUSION - the results are not clearly presented and major revision is recommended.
Response 10: We introduced the corresponding changes to the manuscript.

Round 2
Reviewer 1 Report
Authors have made an interesting effort investigating the regulation of Endothelin-1 and the NO homeostasis during ischemia-reperfusion in terms of a clinically applicable hypoxic/hyperoxic ischemic preconditioning. Authors have modified the manuscript, correcting linguistic and grammar errors in the manuscript. However, despite the comments, authors have not conducted any further molecular analyses and the paper remains mostly descriptive.
Major Comments
- The direct effects of the hypoxic/hyperoxic preconditioning on the myocardium is only investigated through histology. It would be interesting if authors could address the molecular signalling of hypoxic/hyperoxic preconditioning, in terms of eNOS phosphorylation status or it’s upstream effectors Akt and PI3K (Bibli SI et al. J Cardiovasc Pharmacol Ther. 2014 Mar;19(2):220-7). This would facilitate the mechanistic insight between NO release and myocardial damage.
- ET-1 has been shown to disturb Ca2+ homeostasis and induce inotropy and Ca2+-mediated apoptosis in the cardiomyocytes (Vignon-Zellweger N, et al. Life Sci. 2012). It would be therefore interesting is authors could -alongside eNOS signalling- investigate sarcoplasmic reticulum homeostasis through studying key SR molecules regulation.
- The most concerning issue of the manuscript is the translational value of the protocol. Even though preconditioning is proven to be a highly cardioprotective mechanism, its translational value is minimal as in clinical praxis, it is highly impossible to foresee an upcoming acute myocardial infarction and apply a prompt intervention. It would be interesting if authors could investigate the same cardioprotective potential in terms of ischemic postconditioning, which is more clinically relevant.
If authors cannot investigate any of the aforementioned issues, they should at least add the concerns as limitations to the work and include the aforementioned issues in the discussion section.
Author Response
Response to Reviewer 1 Comments
Major Comments
- The direct effects of the hypoxic/hyperoxic preconditioning on the myocardium is only investigated through histology. It would be interesting if authors could address the molecular signalling of hypoxic/hyperoxic preconditioning, in terms of eNOS phosphorylation status or it’s upstream effectors Akt and PI3K (Bibli SI et al. J Cardiovasc Pharmacol Ther. 2014 Mar;19(2):220-7). This would facilitate the mechanistic insight between NO release and myocardial damage.
Response 1: Thank you for your valuable comments and support of our manuscript! We introduced the corresponding changes to the manuscript.
Certainly it would be interesting to address the molecular signalling of preconditioning and to discover the exact mechanism of preconditioning effect. Bibli SI et al. study is similar to ours. We all trying to provide experimental methods of organ protection aiming to help people to survive through critical illness. In our study we investigate the methods of organ protection during cardiosurgery with cardiopulmonary bypass. Cardiopulmonary bypass (CPB) is an inalienable part of the operation during which the ischemia-reperfusion injury occurs inevitably.
The molecular signalling of hypoxic/hyperoxic preconditioning will be the next part of our study. Unfortunately, we have no opportunity to do such investigation on this material. The histology, as basic method of study of tissue structure, shows appropriate difference between preconditioning methods.
- ET-1 has been shown to disturb Ca2+ homeostasis and induce inotropy and Ca2+-mediated apoptosis in the cardiomyocytes (Vignon-Zellweger N, et al. Life Sci. 2012). It would be therefore interesting is authors could -alongside eNOS signalling- investigate sarcoplasmic reticulum homeostasis through studying key SR molecules regulation.
Response 2: Thank you for your valuable comments and support of our manuscript! We introduced the corresponding changes to the manuscript.
Undoubtedly it would be interesting to investigate sarcoplasmic reticulum homeostasis through studying key SR molecules regulation. In our study we observed the vasoconstriction and vasodilatation effects of changes in ET-1 and NO production during stress (ischemia-reperfusion injury) on the background of preconditioning or without it. So we hypothesise that changes in vascular tone are the basis of vascular training before stress situation. We are looking forward to provide more studies (including patients) to establish the exact vascular reaction and molecular signalling of preconditioning in clinical practice.
- The most concerning issue of the manuscript is the translational value of the protocol. Even though preconditioning is proven to be a highly cardioprotective mechanism, its translational value is minimal as in clinical praxis, it is highly impossible to foresee an upcoming acute myocardial infarction and apply a prompt intervention. It would be interesting if authors could investigate the same cardioprotective potential in terms of ischemic postconditioning, which is more clinically relevant.
Response 3: Thank you for your valuable comments and support of our manuscript!
Coronary artery bypass grafting (CABG) with cardiopulmonary bypass is a common surgical therapy for patients suffering from coronary artery diseases. During surgery, the heart is subjected to a long period of ischemia due to the occlusion of aortic artery. The heavy burden of myocardial ischemia-reperfusion injury thus induces cardiomyocyte death, which can paradoxically reduce the beneficial effect of CABG (Qi-Wen Deng, et al. Clinical benefits of aortic cross-clamping versus limb remote ischemic preconditioning in coronary artery bypass grafting with cardiopulmonary bypass: a meta-analysis of randomized controlled trials. Journal of surgical research. 2015, 193; 52-68 doi.org/10.1016/j.jss.2014.10.007)
Acute cardiovascular dysfunction occurs perioperatively in more than 20% of cardiosurgical patients. Twenty-five percent of patients undergoing elective CABG surgery require inotropic support for postoperative myocardial dysfunction. CABG surgery is associated with systemic inflammatory response, endothelial damage and platelet activation regardless of the use of CPB.
Of course it is impossible to foresee an upcoming acute myocardial infarction. The postconditioning is more clinically relevant for this patients.
According to the systematic review and meta-analysis of randomized controlled trials remote ischemic preconditioning did not consistently reduce morbidity and mortality in adults undergoing cardiac surgery with CPB (Giacomo Deferrari et al., Nephrol Dial Transplant (2018) 33: 813–824).
Different preconditioning methods are proven in many studies to be a highly cardioprotective. Its translational value is clinically relevant especially in cardiosurgery. Cardiosurgery patients are often old and have comorbidities. So, the cardio operation is hazardous to the length and quality of life of the patients, because of ischemia-reperfusion injury that is inevitably during CPB. The duration of CPB is depends on the type of the operation and the ability of the surgeon to do all the best as fast as possible. Many operations lasts more then 5 hours with CPB of 60-120 minutes. The systemic inflammatory reaction syndrome is manifested during nonpulsatile mode of CPB. Not only myocardium but other organs are affected with hypoperfusion and inflammation during CBP. The preconditioning methods are able to improve or prevent the organs damage during CPB.
4. If authors cannot investigate any of the aforementioned issues, they should at least add the concerns as limitations to the work and include the aforementioned issues in the discussion section.
Response 4: Thank you for your valuable comments and support of our manuscript! We introduced the corresponding changes to the manuscript.

Reviewer 2 Report
The manuscript by Mandel et al. has been considerably improved. I have no further comments and can recommend this paper for publication.
Author Response
Thank you for your valuable comments and support of our manuscript!
Round 3
Reviewer 1 Report
The authors did not reply in the manuscript for my first comment. Please refer to the similar publication (Bibli SI et al.) with yours, as you state and clearly state as study limitations that "The molecular signalling of hypoxic/hyperoxic preconditioning will be the next part of your study".
Author Response
Minor Comments
- The authors did not reply in the manuscript for my first comment. Please refer to the similar publication (Bibli SI et al.) with yours, as you state and clearly state as study limitations that "The molecular signalling of hypoxic/hyperoxic preconditioning will be the next part of your study".
Response 1: Thank you for your support of our manuscript! We introduced the corresponding changes to the manuscript.
